# Isolation and Characterisation of *Streptococcus* spp. with Human Milk Oligosaccharides Utilization Capacity from Human Milk

**DOI:** 10.3390/foods13091291

**Published:** 2024-04-23

**Authors:** Ye Zhou, Xiaoming Liu, Haiqin Chen, Jianxin Zhao, Hao Zhang, Wei Chen, Bo Yang

**Affiliations:** 1State Key Laboratory of Food Science and Resources, Jiangnan University, Wuxi 214122, China; 6210112137@stu.jiangnan.edu.cn (Y.Z.); liuxm@jiangnan.edu.cn (X.L.); haiqinchen@jiangnan.edu.cn (H.C.); zhaojianxin@jiangnan.edu.cn (J.Z.); zhanghao61@jiangnan.edu.cn (H.Z.); chenwei66@jiangnan.edu.cn (W.C.); 2School of Food Science and Technology, Jiangnan University, Wuxi 214122, China; 3Institute of Food Biotechnology, Jiangnan University, Yangzhou 225004, China; 4National Engineering Research Center for Functional Food, Jiangnan University, Wuxi 214122, China; 5International Joint Research Laboratory for Pharmabiotics & Antibiotic Resistance, Jiangnan University, Wuxi 214122, China

**Keywords:** human milk oligosaccharides (HMO), *Streptococcus* spp., safety-related genes

## Abstract

Human milk oligosaccharides (HMO) that promote the growth of beneficial gut microbes in infants are abundant in human milk. *Streptococcus*, one of the dominant genera in human milk microbiota, is also highly prevalent in the infant gut microbiota, possibly due to its adeptness at utilizing HMOs. While previous studies have mainly focused on HMO interactions with gut bacteria like *Bifidobacterium* and *Bacteroides* spp., the interaction with *Streptococcus* spp. has not been fully explored. In this study, *Streptococcus* spp. was isolated from human milk and identified to exhibit extensive capabilities in utilizing HMOs. Their consumption rates of 2′-fucosyllactose (2′-FL), 6′-sialyllactose (6′-SL), and lacto-*N*-tetraose (LNT) closely matched those of *Bifidobacterium longum* subsp. *infantis* ATCC 15697. Furthermore, we assessed the safety-related genes in the genomes of the *Streptococcus* species capable of utilizing HMOs, revealing potential virulence and resistance genes. In addition, no haemolytic activity was observed. These findings expand the knowledge of metabolic interactions and networks within the microbiota of human milk and the early life human gut.

## 1. Introduction

The infant gut microbiota undergoes extreme fluctuations in membership and overall abundance during the first month of life [1]. However, the factors driving infant microbiota assembly are poorly understood. In recent years, breastfeeding has been considered to play an important role in the formulation of infant gut microbiota [2]. Human milk oligosaccharide (HMO) is the third most abundant molecular species after lactose and lipids and is present at a percentage of 1.0–1.5% in human milk [3]. HMO consists of a lactose core at the reducing end and 1–13 elongating carbohydrate units, including glucose, galactose, *N*-acetylglucosamine, fucose, and *N*-acetylneuraminic acid. Lactose can be linked with fucose through α1-2 or α1-3 bonds to generate 2′-fucosyllactose (2′-FL) or 3-fucosyllactose (3-FL), respectively. In addition, it can be connected with sialic acid through α2-3 or α2-6 bonds to produce 3′-sialyllactose (3′-SL) or 6′-sialyllactose (6′-SL). Furthermore, lactose can be bonded with galactose β-1,3-*N*-acetylglucosamine to form lacto-*N*-tetraose (LNT) or with *N*-acetyllactosamine to create lacto-*N*-neotetraose (LNnT) [4].

Due to their role as prebiotics, HMOs are considered one of the motivations for changing the intestinal microbiota composition [5]. HMOs are fermented in the lower part of the intestine by members of intestinal microbiota, thus multiplying these HMO-utilizing microbes. Among all the members of gut microbiota in breastfeeding infants, *Bifidobacterium* species were previously confirmed to utilize HMO, and their underlying patterns for HMO uptake, degradation, and metabolism have been revealed in depth [6,7,8]. Recently, it was discovered that HMO utilization was not exclusive to the *Bifidobacterium*, since some strains of *Lactobacillus*, *Bacteroides*, *Roseburia*, and *Eubacterium* also showed HMO-utilizing ability [9,10,11].

The *Streptococcus* genus, despite its variable abundance, is commonly present in the infant gut microbiota [12]. The HMO-utilizing capacity of *S. mitis* and *S. oralis* has been demonstrated [13], in which the OD_600_ of the two *Streptococcus* strains grown in medium supplied with2′-FL were over 10% more than those grown in medium without supply. Thus, *Streptococcus* may contribute to the changes in infant gut microbiota by HMO utilization. However, the HMO-utilizing capacity of other *Streptococcus* species, as well as the HMO types they ferment, remains to be revealed. Additionally, the majority of *Streptococcus* strains are potential pathogens, and the safety of those capable of utilizing HMO remains to be further elucidated.

In the current study, the ability of *Streptococcus* strains, isolated from the human milk of healthy women, to ferment HMOs including 2′-FL, 3′-FL, 3′-SL, 6′-SL, LNT, and LNnT was confirmed. Additionally, genomic analysis that was suggestive of the presence of putative safety genes in *Streptococcus* strains was performed. These findings may help to elucidate the fluctuations in membership and the overall abundance of *Streptococcus* in gut microbiota during infancy, as well as reveal the safety implications of *Streptococcus* strains capable of utilizing HMOs.

## 2. Materials and Methods

### 2.1. Isolation and Cultivation of the Strains

The human milk samples of healthy women were collected in the Affiliated Wuxi Maternity and Child Health Care Hospital of Nanjing Medical University and stored at −80 °C before use. An aliquot of 100 μL that was serially diluted in saline (1:10 in volume) was incubated on modified MRS agar (mMRS agar, containing 10 g tryptone, 20 glucose, 10 g beef extract, 5 g yeast extract, 2 g K_2_HPO_4_, 2 g ammonium citrate, 2 g sodium acetate, 0.5 g MgSO_4_, 0.25 g MnSO_4_, 0.5 g L-cysteine, 1 mL Tween 80, and 20 g agar per litre) containing 5 g/L glucose under strictly anaerobic conditions at 37 °C for 48 h. The colonies were selected randomly and streaked and purified twice on mMRS agar and then sub-cultured twice in mMRS broth under the same conditions before use. The cultivation steps were conducted using an anaerobic incubation system (AW500SG, Don Whitley Scientific, Bingley, UK) containing 85% N_2_, 10% H_2_, and 5% CO_2_.

### 2.2. Taxonomic Identification 

The 16S rRNA gene amplification of isolated strains was conducted as previously described [14], and sequencing was performed by GENEWIZ Co., Ltd. (Suzhou, China). The sequences were blasted based on the NCBI database and strains primarily identified as *Streptococcus* were selected for further research. 

### 2.3. HMO Fermentation and Growth Curve 

The *Streptococcus* strains were cultured in M17 broth, which consists of 5 g tryptone, 5 g soya peptone, 5 g beef extract, 2.5 g yeast extract, 0.12 g MgSO_4_, 0.5 g ascorbic acid, 19 g sodium β-glycerophosphate, and 5 g glucose per litre, within a constant temperature incubator set at 37 °C for 24 h. HMO fermentation was tested in the M17 medium containing 5 g/L glucose, galacto-oligosaccharides (GOS), 2′-FL, 3′-FL, 3′-SL, 6′-SL, LNT, or LNnT as the sole carbohydrate and 75 mg/L bromocresol purple as a colour indicator. Bromocresol purple was an acid–base indicator that shifted from purple at a pH of 6.8 to yellow at a pH of 5.2. The strains with HMO-utilizing capacity produce an acid, resulting in a pH reduction and a consequent colour transition of the indicator from purple to yellow. Based on the previous conclusions [15], *B. longum* subsp. *infantis* ATCC 15697 was included as a positive control for HMO fermentation. The resulting culture of 2 μL was used to inoculate 200 μL of various media distributed in the wells of a 96-well microplate and incubated in an anaerobic incubator (AW500SG, Don Whitley Scientific, Bingley, UK) flushed with 85% N_2_, 10% H_2_, and 5% CO_2_ for 24 h. The HMO utilization of the strains was identified through the colour change in the corresponding medium. The 2′-FL, 3′-FL, 3′-SL, 6′-SL, LNT, and LNnT used in the study were provided by DSM-Firmenich, and the purities were 98.9%, 92.4%, 95.2%, 98.8%, 86.62%, and 99.4%, respectively.

In the growth curve analysis, *Streptococcus* strains were cultured in M17 broth in a constant temperature incubator at 37 °C until reaching an OD_600_ of 0.6–0.8. A 1% (*v*/*v*) resulting culture was inoculated into 10 mL M17 medium with 2′-FL, 6′-SL, or LNT as the sole carbohydrate and cultured for 24 h at 37 °C. GOS or glucose was used as the positive control, and the sugar-free medium was used as the negative control. Cell growth was measured every 2 h by taking optical density measurements at a wavelength of 600 nm on a spectrophotometer (Ultrospec 1100 pro, Amersham Biosciences, Pittsburgh, PA, USA). The maximum growth rate was calculated through fitting the growth curves of *Streptococcus* spp. with a logistic model [16]. The pH curve was generated by measuring the pH value of the culture at the same frequency.

### 2.4. HMO Consumption and Metabolite Quantitation

The *Streptococcus* strains were cultured in M17 media supplemented with 5 g/L of 2′-FL, 6′-SL, or LNT as the sole carbon source for either 10 or 24 h. Fermentation supernatants collected at these distinct time points were added with 200 μL of chloroform–n-butanol (4:1) solvent per 1 mL supernatant, followed by centrifugation at 12,000× *g* for 10 min to precipitate proteins. The supernatant was diluted with deionized water to proper concentration and centrifuged at 8000× *g* for 5 min to obtain the supernatant for further analysis. The HMO detection method was referenced from previous description with appropriate modifications [17]. Specifically, an HPLC–MS system (Thermo Fisher Scientific, Waltham, Massachusetts, USA) with a Waters ACQUITY UPLC^®^ BEH Amide column (2.1 mm × 100 mm) was used for HMO quantitation. Mobile phases A and B were composed of a 25 mM ammonium acetate solution and acetonitrile, respectively. The flow rate was 0.3 mL/min, and the column temperature was at 15 °C. The optimized gradient was 85% solution B (acetonitrile) at the initial 1 min, 85–65% solution B for 1.00–14.00 min, 65–40% solution B for 14.00–16.00 min, 40% solution B for 16.00–18.00 min, 40–85% solution B for 18.00–18.10 min, and 85% solution B for 18.10–23.00 min.

For the quantification of acetic acid, lactic acid, and 1,2-propanediol, *Streptococcus* strains were cultured in M17 media with glucose, 2′-FL, 6′-SL, or LNT (each at a concentration of 5 g/L) as the sole carbon source for 24 h. The supernatant at the end of the fermentation was collected and processed according to the aforementioned method. The high-performance liquid chromatography (HPLC) system (Thermo Fisher Scientific, MA, USA) was used, equipped with a refractive index detector (RID) and an Aminex HPX-87H column (300 mm × 7.8 mm) maintained at 50 °C. Elution was carried out with 5 mM H_2_SO_4_ solution at a consistent flow rate of 0.5 mL/min.

### 2.5. Genome Mining Analysis

The bacterial genomic DNA were extracted with a TIANamp Bacteria DNA Kit (TianGen Co., Ltd., Beijing, China) according to the manual. Genomic DNA was sequenced and annotated by Majorbio (Shanghai, China). Briefly, the Illumina Hiseq × 10 platform was used to sequence the draft genomes, which generated 2 × 150 bp paired-end libraries with an average read length of approximately 400 bp. Quality trimming and the filtering of Illumina reads were performed, including trimming the 5′ ends to ensure that base abundance per position fell within two standard deviations of the average across all cycles. Bases with a quality score below 20 at the 3′ ends were trimmed, and reads shorter than 35 bases or with a median quality score below 20 were removed. SOAPdenovo v2.04 was used to assemble the reads and GapCloser v1.12 was used to fill the local inner gaps, referring to previous research [18]. The GC depth distribution and K-mer distribution were used to evaluate the quality.

The Virulence Factor Database (VFDB) (https://card.mcmaster.ca/ (accessed on 11 March 2024)) [19] served to identify potential virulence factors within the genomes of the *Streptococcus* strains, with criteria for identification set at a minimum of 70% sequence identity and an e-value threshold of less than 0.01. Prophages were predicted using the PHASTER database (http://phaster.ca/ (accessed on 11 March 2024)) [20], and the total intact prophage number, length, and GC content were quantified for each bacterial strain. The Comprehensive Antibiotic Resistance Database (CARD; https://card.mcmaster.ca/ (accessed on 11 March 2024)) [21] was employed for the prediction of genes associated with antibiotic resistance.

### 2.6. Haemolysis Activity

To assess the haemolytic activity of the *Streptococcus* strains, the organisms were cultured on Columbia blood agar plates containing 5% sheep blood and incubated at 37 °C for 48 h. The presence of haemolytic activity was ascertained by observing any alteration in colour in the regions directly adjacent to the colonies of the bacteria.

## 3. Results

### 3.1. Isolation of Streptococcus spp. with HMO Utilization Capability

A total of 46 strains of *Streptococcus* were isolated from the human milk of a healthy mother. According to the 16S rRNA sequences, 42 strains were most closely related with *S. salivarius*, and 1 strain was related with *S. mitis*. The other 3 strains were not attributed to any known *Streptococcus* species. The 16S rRNA sequences of these 4 *Streptococcus* sp. strains are presented in Appendix A. The various isolated strains were evaluated for their abilities to utilize 2′-FL,3′-FL, 3′-SL, 6′-SL, LNT, and LNnT (Figure 1), and bromocresol purple was used as the indicator. The change in colour from purple into yellow in the medium was observed for all strains after growth over 24 h with 2′-FL,3′-FL, 3′-SL, 6′-SL, LNT, or LNnT (Figure 1), except for *S. salivarius* strains. The colour changed in the medium indicated that *Streptococcus* sp. 21WXBC0057M1, 21WXBC0044M1, BJSWXB5TM5, and JSWX21MRM4 exhibited the ability to utilize 2′-FL,3′-FL, 3′-SL, 6′-SL, LNT, and LNnT, whereas *S. salivarius* strains showed no capability to utilize any HMOs. Given that 2′FL, 6′SL, and LNT represent the predominant structural types of HMOs in most women’s breast milk, we selected these compounds to further investigate the HMO utilization characteristics of the four *Streptococcus* sp. strains.

To precisely evaluate the utilization capability of HMO, four *Streptococcus* sp. strains were selected for growth assessment on 2′-FL, 6′-SL, or LNT, with glucose and GOS as positive controls. Overall, all four bacterial strains exhibited notable growth after 24 h on media containing 2′-FL, 6′-SL, or LNT (Figure 2). Nonetheless, the maximum biomass of *Streptococcus* on HMO as the sole carbohydrate was inferior to that on glucose, with the exception of strains 21WXBC0044M1 and JSWX21MRM4 when they were cultivated on 2′-FL and LNT medium. Distinct maximum growth rates were also observed in *Streptococcus* growth on glucose and HMO (Figure 2). In particular, 21WXBC0057M1 exhibited a growth rate of 2.0 h^−1^ in 6′-SL, while achieving a rate of 3.1 h^−1^ in glucose. JSWX21MRM4 strain grew efficiently when fed on 2′-FL, 6′-SL, or LNT, which was consistent with previous observations.

### 3.2. Characteristics of HMO Utilization by Streptococcus

It was previously confirmed that *Streptococcus* strains were capable of utilizing HMO; however, their utilization efficiency, especially when compared to that of *Bifidobacteria*, had yet to be quantitatively determined. Using the method of HPLC–MS quantification, we proceeded to evaluate the HMO consumption of these four *Streptococcus* strains, along with *B. longum* subsp. *infantis* ATCC15697, a typical strain with expansive HMO consumption capabilities, as a reference strain. The glycoprofiles based on 2′-FL, 6′-SL, or LNT showed that both *Streptococcus* spp. and *B. longum* subsp. *infantis* ATCC15697 had a low HMO depletion rate (<15%) within 10 h, except for 21WXBC0044M1, which consumed 31%, 16%, and 29% of 6′-SL, 2′-FL, and LNT, respectively (Figure 3). However, after 24 h of cultivation, *Streptococcus* spp. consumed more HMO with only negligible amounts remaining. The HMO decreases in *Streptococcus* spp. were substantially higher than that of *B. longum* subsp. *infantis* ATCC 15697 (Figure 3).

During bacterial growth, 2′-FL, 6′-SL, and LNT have been shown to be ultimately metabolized into lactic acid, acetic acid, or 1,2-propanediol (1,2-PD) by *B. longum* subsp. *infantis* [22]. We interrogated the certain types and amounts of end-products produced by *Streptococcus* spp. during the cell growth. A higher level of acetate was produced by the strains cultivated on HMO (Figure 4), particularly on 6′-SL and LNT in comparison with that on glucose. These noticeable increases in acetate were interpreted to be generated by *Streptococcus* spp. acting on other HMO constituents apart from glucose and galactose (transformed by bacteria into glucose for metabolism). This was supported by a previous study indicating that fucose, sialic acid, and *N*-acetylglucosamine, three constituent unique monosaccharides of HMO, could be further degraded by *B. longum* subsp. *infantis* and converted into lactate and acetate via the bifid shunt [22]. In addition, the production of acetic and lactic acid by the *Streptococcus* strains resulted in a significant pH reduction in the culture medium from 7.2 to less than 5.0 (Figure 5). The pH curve was also negatively correlated with the growth curve of the four strains, indicating their simultaneous growth and fermentation. However, 1,2-propanediol, one of the common terminal metabolites transmitted from fucose residues in 2′-FL [23], was absent in the fermentation supernatants of the four *Streptococcus* strains (Figure 4). This marked a significant difference in the HMO metabolic profile between *Streptococcus* spp. and *B. longum* subsp. *infantis* ATCC 15697, indicating the unique 2′-FL metabolic pathway that *Streptococcus* strains engages in. This may be due to the unique metabolic pathway for *Streptococcus* spp. to 2′-FL, which might be substantially different between *Streptococcus* and *Bifidobacterium*.

### 3.3. In Silico and In Vitro Safety Assessment of the Three Undefined Streptococci

Among the four strains with extensive HMO utilization ability above, the strains 21WXBC0057M1, 21WXBC0044M1, and BJSWXB5TM5 were not attributed to any known *Streptococcus* strains according to the 16S rDNA sequences. Thus, we conducted a safety gene analysis on the genomes of these three undefined *Streptococcus* strains. The draft genomes of these three strains have been uploaded to NCBI, with the GenBank accession numbers being JAXHDO000000000 (21WXBC0057M1), JAXHDN000000000 (21WXBC0044M1), and JAXHDP000000000 (BJSWXB5TM5T), respectively.

Virulence factors are essential molecules that facilitate bacterial colonization within host cells. These factors encompass invasion, mucin degradation, cytotoxicity, haemolysis, and biofilm formation, all of which are critical to bacterial pathogenicity [24]. The virulence Factors Database (VFDB) was used to identify the classic virulence factors in the genomes. The analysis based on the full database revealed the potential virulence factors genes within the genomes of three *Streptococcus* strains (Table 1). *Psa*A, *pav*A, and *has*C were commonly present among the three strains, and unique factors were observed in each strain: a Capsule in 21WXBC0057M1, PI-2 and Capsule in 21WXBC0044M1, and Neuraminidase in BJSWXB5TM5.

Prophages represent another type of mobile genetic element in bacteria that integrates into the host genome, containing functional genes that enhance bacterial adaptability to the environment [25]. Using PHASTER for prophage analysis on the three *Streptococcus* strains, intact prophage sequences were discovered in strains 21WXBC0057M1 and BJSWXB5TM5 (Table 2). These sequences exhibit notable similarities to the coding sequences (CDS) of known streptococcal phages, specifically PH15 and phi3396. In contrast, strain 21WXBC0044M1 lacked intact prophage sequences, highlighting the genetic diversity and complexity among these *Streptococcus* strains.

Antimicrobial resistance has arisen as a vital worldwide threat to human health. Potential antimicrobial resistance could be indicated by the corresponding gene alignment [26]. Thus, the prediction of antibiotic resistance genes in bacteria is a pivotal facet of safety evaluation. The CARD was used to predict potential resistance genes in the genomes of three *Streptococcus* strains. A total of 68 antibiotic resistance genes were discovered in the genomes of the three strains, with *mac*B, *tet*A(58), *bcr*A, and *ole*C being the most prevalent (Figure 6). These genes are responsible for encoding membrane-bound efflux pumps that actively expel antibiotics from the bacterial cells, thus safeguarding the bacteria from the lethal effects of macrolides, tetracycline, bacitracin, or oleandomycin. Furthermore, genes conferring resistance to vancomycin, including *van*RE, *van*RM, *van*SB, *van*TG, *van*XYG, and *van*B, were consistently detected across all three strains. These genes are deemed intrinsic and incapable of horizontal gene transfer [27]. In addition, resistance genes for aminocoumarin (*par*Y), bacitracin (*bac*A), and fosfomycin (*mur*A) were also discovered in these *Streptococcus* strains.

The absence of haemolytic activity is a critical criterion established by FAO/WHO (2002) for the appraisal of potential probiotic bacteria [28]. Both α and β haemolytic activities are considered unfavourable for probiotic candidacy. The *Streptococcus* encompasses species with strongly haemolytic and pathogenic characteristics, including *S. pneumoniae* and *S. pyogenes*. Given that strains 21WXBC0057M1, 21WXBC0044M1, and BJSWXB5TM5 were undefined species within the *Streptococcus*, it is imperative to evaluate their haemolytic potential. Our findings showed an absence of haemolytic circles for these strains when cultured on blood agar plates (Figure 7), indicating that all of them exhibited γ-haemolytic activity in vitro. γ-haemolysis, which signifies non-haemolytic activity, does not induce the rupture of red blood cells. Lactic acid bacteria isolated from curd samples were found to exhibit γ-haemolytic activity [29].

## 4. Discussion

Emerging evidence highlights the critical role of HMOs in the development of the gut microbiome and the immune system during early life. The health benefits of HMO for infants are intricately linked to their absorption and metabolic processing by gut bacteria, with consequential impacts on the infant’s gut microbiome and immune function. *Streptococcus* spp. are crucial parts of both human milk and early life gut microbiota, yet little is known regarding their utilization of HMO. Three *Streptococcus* strains (21WXBC0057M1, 21WXBC0044M1, and BJSWXB5TM5T) isolated from the human milk of healthy mothers, along with the JSWX21MRM4 strain, demonstrated an extensive ability to metabolize HMOs, including 2′-FL, 3′-FL, 3′-SL, 6′-SL, LNT, and LNnT. This study provided additional HMO-utilizing *Streptococcus* strains following *S. mitis* and *S. oralis*, focusing on their growth phenotype, HMO depletion rate, metabolites, and safety profiles.

The growth of bacteria in certain carbon source is the most direct evidence to identify their carbohydrate utilization [30,31]. Research on bacterial growth has revealed that a limited number of gut bacteria, mainly *Bifidobacteria* and specific *Bacteroides* strains, grew rapidly with HMO as the sole carbon source [9,32]. The growth profiles in the current study revealed that *Streptococcus* spp. strains could survive with HMO as their only carbon source. This was in good agreement with previous research, showing that fucosylated HMOs notably provided a growth advantage to *S. mitis* [13]. Notably, *Streptococcus* spp. exhibited more efficient and rapid growth on glucose and GOS compared to HMO. The growth rate of *Streptococcus* spp. with HMO was not as fast as that with glucose, the trends of which are similar to those reported in *B. longum* subsp. *infantis* ATCC 15697 and *Bacteroides fragilis* ATCC 25285 [33].

Carbohydrate consumption has been widely used to quantify the utilization capability of strains [34]. Aligned with the growth metrics, *Streptococcus* spp. was an efficient HMO utilizer, with negligible HMO remaining in the cell-free supernatant at the 24th hour. Of note, *Streptococcus* spp. exhibited a superior rate of HMO consumption relative to that of *B. longum* subsp. *infantis* ATCC 15697 within 10 h. In addition to the different HMO utilization capabilities, this may also be attributed to the differential growth kinetics between *Streptococcus* and *Bifidobacterium*, as *B. longum* subsp. *infantis* ATCC 15697 typically entered the logarithmic growth phase after 16 h of fermentation (Appendix A). Moreover, the biomass and maximum growth rate of *B. longum* subsp. *infantis* ATCC 15697 obtained on HMO medium were inferior to those of *Streptococcus* spp. on HMO medium (Appendix A). Thus, it could be inferred that the above-mentioned *Streptococcus* may possess enhanced HMO-utilizing capability compared to *B. longum* subsp. *infantis* ATCC 15697.

HMOs are usually converted by bacteria into organic acids, such as lactic acid and short-chain fatty acids [35]. In the current study, substantial concentrations of lactic acid and acetic acid were observed in the supernatants of HMO fermentation by *Streptococcus* spp., whereas 1,2-propanediol remained undetectable. This absence may be attributed to the lack of a fucose metabolic pathway [36] in *Streptococcus* spp. The generation of lactic acid and acetic acid during HMO degradation decreases the pH in the intestinal tract and establishes an acidic environment, thus hindering the growth of pathogens and maintaining intestinal health [37].

It is universally recognized that the *Streptococcus* encompasses microbes with inherent pathogenic potential [38]. In particular, the capacity to metabolize a wide range of carbohydrates is reported as a pivotal factor for the virulence of pathogenic streptococci, possibly facilitating their sustenance within specific ecological niches [39,40]. Thus, great necessity should be attached to the safety assessment of the *Streptococcus* strains. We conducted the genomic virulence factor assessment of these strains, and several genes including *psa*A, *pav*A, and *has*C were discovered in 21WXBC0057M1, 21WXBC0044M1, and BJSWXB5TM5 strains. These genes exhibited more than 70% homology with those of *Streptococcus pneumoniae* and *Streptococcus pyogenes* and have been confirmed to be involved in adhesion, binding to adhesion lipoproteins, and inhibiting phagocytosis [39,41]. However, these genes were not interpreted as negative, for certain members of them did not confer pathogenicity but instead encoded proteins with essential cellular functions [42,43]. Therefore, the specific functions of the proteins they encoded need further research on the phenotype level.

Antibiotic resistance is another significant aspect of pathogen virulence. In the analysis of three undefined *Streptococcus* strains, *mac*B and *tet*A emerged as the most prevalent antibiotic resistance genes, mediating macrolide and tetracycline efflux, respectively. Of note, several isolates of *Streptococcus salivarius* from healthy infants’ oral cavities had been found to carry *mac*B genes, while *Streptococcus thermophilus* harboured the tetracycline resistance gene *tet*M; however, no transfer of the monitored antibiotic resistance genes was detected [44,45]. Moreover, diverse antibiotic resistance profiles had been identified in lactic acid bacteria [46], with the resistance of probiotics to antibiotics deemed beneficial for their colonization in the gastrointestinal tract [47]. Moreover, 21WXBC0057M1, 21WXBC0044M1, and BJSWXB5TM5 strains were not found to possess any haemolytic activity. Further in vitro and in vivo studies are required to comprehensively assess and validate the safety of these *Streptococcus* strains and to gain insight into their roles in breast milk and the infant gut.

## 5. Conclusions

The current study provided new insights into the HMO utilization abilities and safety profiles of *Streptococcus* strains found in healthy mother’s milk. These strains may be transferred alongside HMO into the infant’s gastrointestinal tract, contributing to the colonization and development of the early life microbiota. Nonetheless, further investigation is imperative to comprehensively evaluate their safety profiles as well as reveal their molecular biological mechanisms for HMO utilization.

## Figures and Tables

**Figure 1 foods-13-01291-f001:**
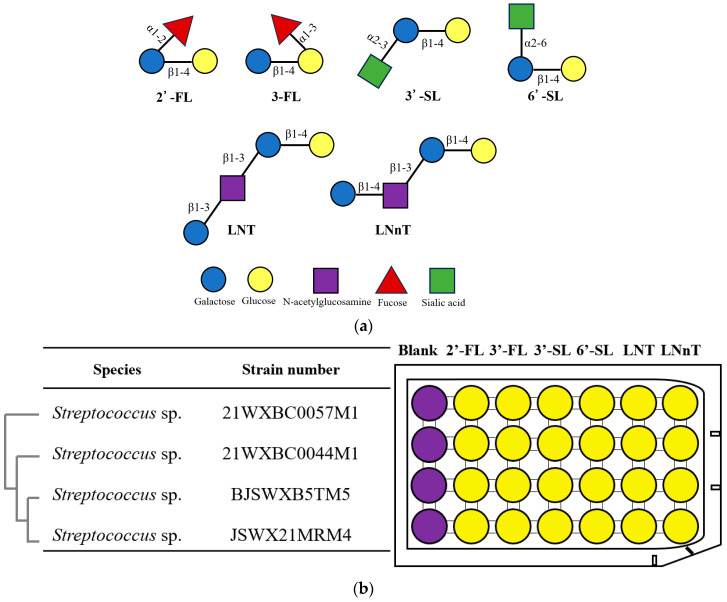
The structure of HMOs and the selection of *Streptococcus* strains. (**a**) HMO structures. (**b**) *Streptococcus* sp. strains with the capability to utilize HMO. The dendrogram conducted with Molecular Evolutionary Genetics Analysis 11 (MEGA 11.0) illustrated the genetic relationships among these *Streptococcus* strains based on the 16S rRNA sequence similarities. The ability of these *Streptococcus* strains to metabolize HMO was evaluated in a 96-well plate, using sugar-free medium as the negative control and bromocresol purple as the metabolic indicator. Purple denoted a lack of metabolic activity, and yellow demonstrated HMO utilization. The carbohydrates are denoted by their abbreviation. 2′-fucosyllactose (2′-FL), 3’-fucosyllactose (3’-FL), 3′-sialyllactose (3′-SL), 6′-sialyllactose (6′-SL), lacto-*N*-tetraose (LNT), and lacto-*N*-neotetraose (LNnT).

**Figure 2 foods-13-01291-f002:**
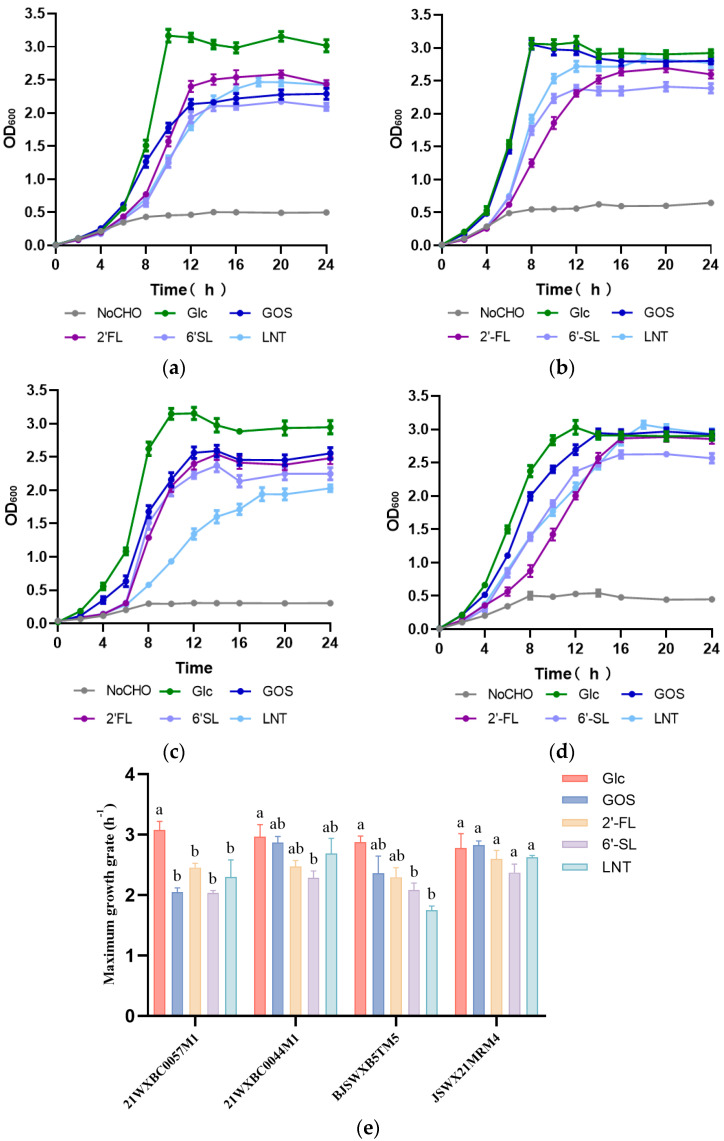
Growth curves and maximum growth rates of *Streptococcus* strains. (**a**) *Streptococcus* sp. 21WXBC0057M1; (**b**) *Streptococcus* sp. 21WXBC0044M1; (**c**) *Streptococcus* sp. BJSWXB5TM5; (**d**) *Streptococcus* sp. JSWX21MRM4. (**e**) Maximum growth rate of *Streptococcus* spp. The carbohydrates are denoted by their abbreviation. No carbohydrate (NoCHO); glucose (Glc). Each data point reflects the average of three replicates, with the error bars denoting standard deviation. The letters a and b indicated statistically significant variances among groups distinguished by different letters (*p* < 0.05).

**Figure 3 foods-13-01291-f003:**
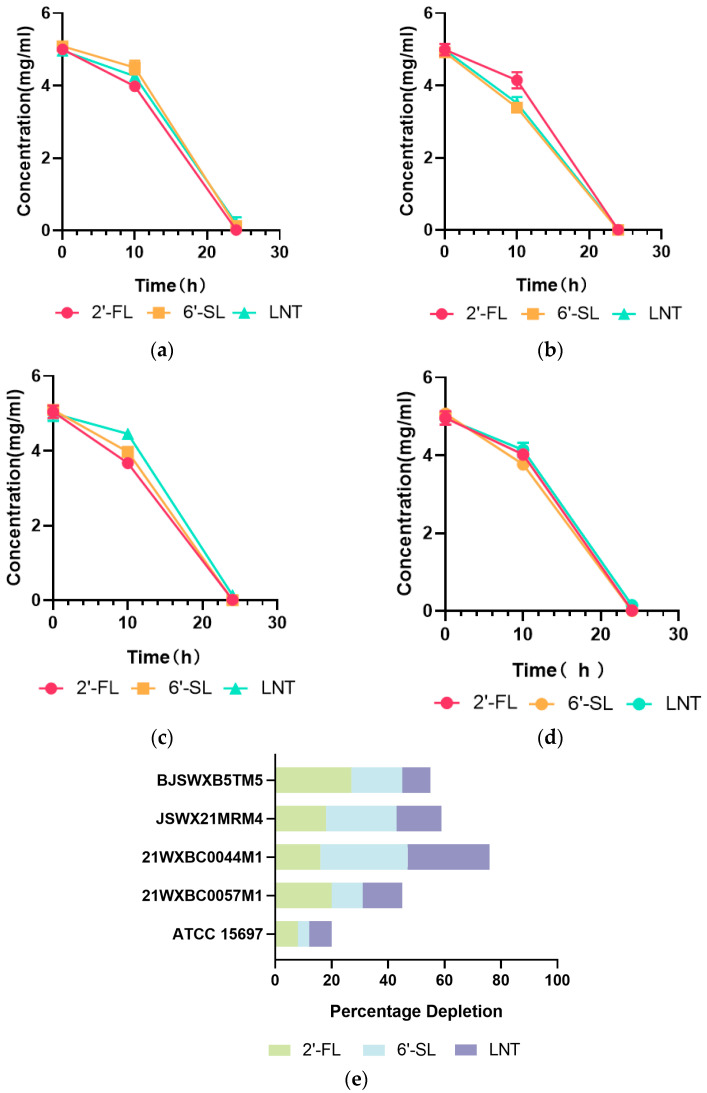
HMO consumption profiles of *Streptococcus* spp. strains. (**a**–**d**) Residual HMO in the culture medium following the cultivation of *Streptococcus* spp. on single HMO (2′-FL, 6′-SL, and LNT at a concentration of 5 g/L) for durations of 10 or 24 h; (**a**) *Streptococcus* sp. 21WXBC0057M1; (**b**) *Streptococcus* sp. 21WXBC0044M1; (**c**) *Streptococcus* sp. BJSWXB5TM5; (**d**) *Streptococcus* sp. JSWX21MRM4. (**e**) Depletion percentages of HMO by *Streptococcus* spp. strains after 10 h of growth in M17 medium, supplemented with 0.5% (*wt.*/*vol*) of 2′-FL, 6′-SL, or LNT. For comparison, the control strain *B. longum* subsp. *infantis* ATCC15697 was cultured in mMRS medium. Triplet incubations were performed for every strain, and the standard deviation in depletion was represented by error bars.

**Figure 4 foods-13-01291-f004:**
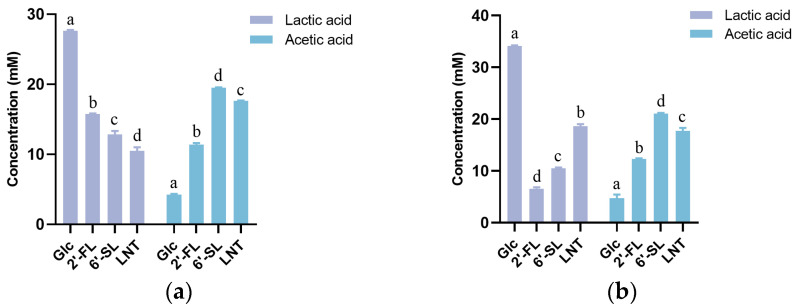
Lactic acid, acetic acid, and 1,2-propanediol produced by *Streptococcus* spp. strains and *B. longum* subsp. *infantis* ATCC15697 when cultured in HMO or glucose; (**a**) *Streptococcus* sp. 21WXBC0057M1; (**b**) *Streptococcus* sp. 21WXBC0044M1; (**c**) *Streptococcus* sp. BJSWXB5TM5; (**d**) *Streptococcus* sp. JSWX21MRM4; (**e**) *B. longum* subsp. *infantis* ATCC 15697. Each strain was cultured in triplicate in M17 medium containing 2′-FL, 6′-SL, or LNT (each HMO at a concentration of 5 g/L). Lactic acid, acetic acid, and 1,2-propanediol were measured after 24 h (48 h for *B. longum* subsp. *infantis* ATCC 15697) of cultivation and adjusted by subtracting the baseline values recorded in a sugar-free medium. The letters a, b, c, and d indicated statistically significant variances among groups distinguished by different letters (*p* < 0.05).

**Figure 5 foods-13-01291-f005:**
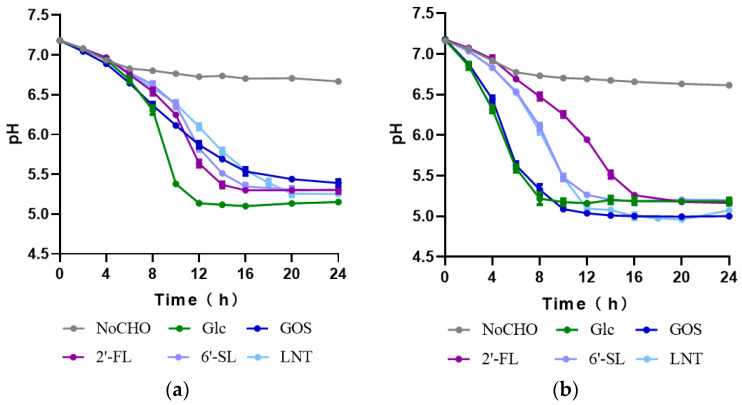
Change in pH due to carbohydrate fermentation by *Streptococcus* spp. (**a**) *Streptococcus* sp. 21WXBC0057M1; (**b**) *Streptococcus* sp. 21WXBC0044M1; (**c**) *Streptococcus* sp. BJSWXB5TM5; (**d**) *Streptococcus* sp. JSWX21MRM4. Each data point reflects the average of three replicates, with the error bars denoting standard deviation.

**Figure 6 foods-13-01291-f006:**
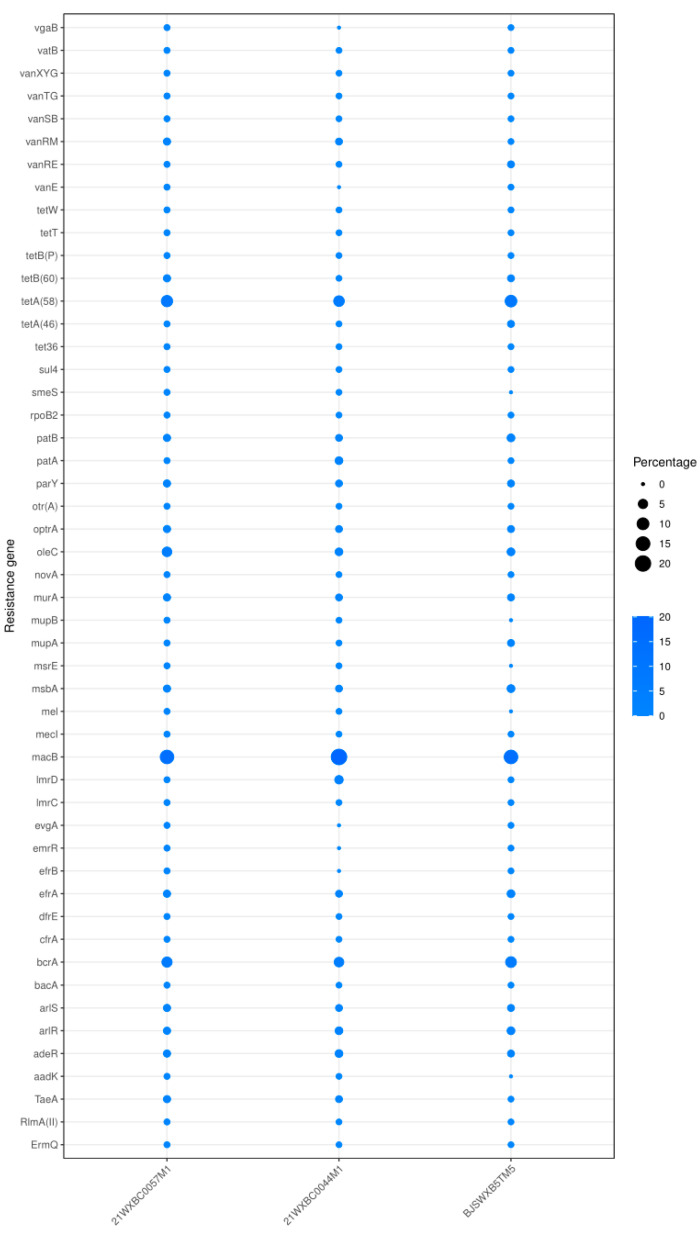
Prediction of resistance genes in *Streptococcus* spp. strains. The size of the circle represents the number of genes.

**Figure 7 foods-13-01291-f007:**
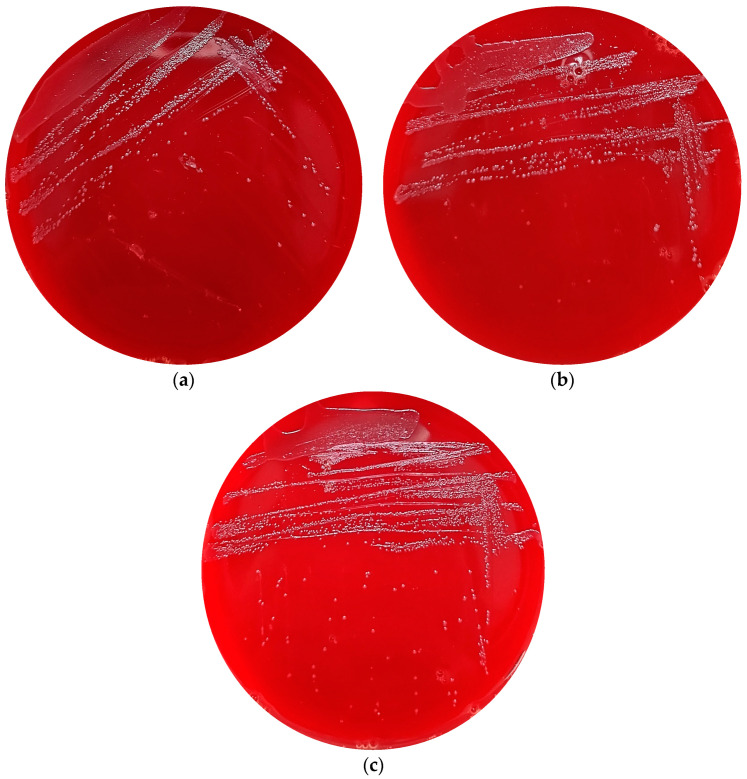
Haemolytic activity of *Streptococcus* sp. strains; (**a**) *Streptococcus* sp. 21WXBC0057M1; (**b**) *Streptococcus* sp. 21WXBC0044M1; (**c**) *Streptococcus* sp. BJSWXB5TM5.

**Table 1 foods-13-01291-t001:** The putative virulence genes in the genome of *Streptococcus* sp. 21WXBC0057M1, 21WXBC0044M1, and BJSWXB5TM5. The positive outcomes were cut off based on a nucleotide identity threshold of 70% and an e-value below 0.01.

Strain	VFDB ID	Gene Name	VFs	Identity (%)	Description
21WXBC0057M1	VFG001359	*psa*A	PsaA	91	Manganese ABC transporter
	VFG005197	*pav*A	PavA	89.4	Adherence and virulence protein
	VFG000964	*has*C	Hyaluronic acid capsule	86.8	Prevents phagocytosis
	VFG001365-VFG001368	*cps4*A, *cps4*B, *cps*C, *cps4*D	Capsule	73.5–74.7	Resistant to complement deposition
21WXBC0044M1	VFG042972 VFG042974	*sip*A, *srt*G1	PI-2	>87.7	Mediates host cell adhesion
	VFG005197	*pav*A	PavA	94.2	Adherence and virulence protein
	VFG001359	*psa*A	PsaA	93.2	Manganese ABC transporter
	VFG001365- VFG001368	*cps4*A, *cps4*B, *cps*C, *cps4*D	Capsule	86.4	Resistant to complement
	VFG000964	*has*C	Hyaluronic acid capsule	79.1	Prevents phagocytosis
	VFG048830	*gnd*A	Capsule	70.6	Protects bacteria from opsonophagocytosis
BJSWXB5TM5	VFG001359	*psa*A	PsaA	93.5	Manganese ABC transporter
	VFG005197	*pav*A	PavA	89.4	Adherence and virulence protein
	VFG000964	*has*C	Hyaluronic acid capsule	86.5	Prevents phagocytosis
	VFG001378	/	Neuraminidase	70	Contributes to increased adhesion

**Table 2 foods-13-01291-t002:** The distribution of intact prophage regions within *Streptococcus* sp. 21WXBC0057M1 and BJSWXB5TM5.

Strain Number	Length	Gene Numbers	The Most Common Prophage	GC Content (%)
21WXBC0057M1	32.1Kb	140	PHAGE_*Strept*_PH15_NC_010945(11)	41.42
BJSWXB5TM5	58.7Kb	140	PHAGE_*Strept*_phi3396_NC_009018(10)	40.25

## Data Availability

The original contributions presented in the study are included in the article/Appendix A, further inquiries can be directed to the corresponding author.

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
