# Peer review of "Isolation and Characterisation of Streptococcus spp. with Human Milk Oligosaccharides Utilization Capacity from Human Milk"

_foods, 2024, doi:10.3390/foods13091291_

Round 1
Reviewer 1 Report
Comments and Suggestions for Authors
This manuscript aimed the isolation and characterization of Streptococcus spp. with HMOs 2 utilization capacity from human milk. The study is of interest to the field of milk microbiology. The experimental work is in general performed well and results are clearly commented. However, some clarifications and changes are necessary before the manuscript can be accepted for publication.
Specific commentsDo not begin a sentence with a number, use for example: An aliquot of.... (revise lines 69, 79 and 91
Explain how the anaerobic condition was obtained in the microplate for the HMOs Fermentation and Growth Curve study.
Add the basis of color change for the HMOs Fermentation and Growth Curve study.
The biomass obtained with glucose was higher than for the rest if the systems, as it was mentioned in the discussion. Revise this point in the results section.
Figure 4: add significant differences between bars.
Comments on the Quality of English LanguageNo comments.
Reviewer 2 Report
Comments and Suggestions for Authors
Review to the manuscript Isolation and characterisation of Streptococcus spp. with HMOs utilization capacity from human milk by Ye Zhou et al.
The study highlights the importance of streptococci (members of Streptococcus genus) presence in human milk and their metabolic potency to consume human milk oligosaccharides (HMOs). These data reveal the importance of the crosstalk between streptococci that originate from human milk and colonize infant gut with the HMOs. In the study, the authors identified three new strains of Streptococcus from human milk capable of HMO species utilization and analysed the genomes to assess the potential safety issues.
The manuscript is generally adequately composed, well-structured and the information clearly presented and discussed. There are clarifications and questions to be addressed, especially on regarding taxonomic issues. Please see specific remarks below.
1) Introduction. The information regarding the linkage types of most common HMO species (referred to and used in the study) should be explained. If appropriate, a structural representation on a scheme could be presented.
2) L 41. Currently the use of the term “microflora” is not encouraged. “Microbiota” is considered more up-to-date.
3) L 46-48. The sentence is not clear and should be rephrased.
4) L49. Are the authors referring to Streptococcus genus or Streptococcaceae family? The taxonomy rank should be specified.
5) L 51-52. Was the growth rate or final OD increased?
6) L 81. The DNA sequences of 16S rRNA genes should be provided as database accessions or in Supplementary.
7) L 88. The origin/provider (and purity) of the used oligosaccahrides should be presented.
8) L 90. Please correct the species name.
9) L 111-112. Which gradient of A and B mobile phase solution was used? If the method is previously used then it should be referenced.
10) Genome mining analysis. The methodology regarding the isolation of genomic DNA, sequencing platform selection, sequencing depth and the bioinformatic analysis is missing. If the genome NGS was ordered as a service, the provider should be specified.
11) L 132, 141 and elsewhere. Why the authors define the strains as completely new species? The information regarding the differences on genomic level from existing species should be presented. There are specific minimal standards set for defining a new species within also Streptococcus genus https://doi.org/10.1099/ijs.0.060046-0. These characteristics are not provided in the paper. Therefore, the isolated streptococci are strains or possibly candidates of novel species, not proved and evaluated as sp. nov. Please note that the species definition and taxonomic rules of prokaryotes has been updated https://doi.org/10.1099/ijsem.0.005585.
12) L139-141. The 16S rDNA sequence identity only is not a valid proof of the correct classification of the species level. If the identity is very high it shows the most probable species. Additional tests should be provided to have reliable confirmation, which was not the case in the manuscript.
13) L 166-168. It is not recommended to use wording such as “less low”, “least slowly”. Please provide the max. growth rates to compare.
14) L 185-186. Were the growth rates of streptococci and B. longum on the used conditions highly similar or not? As the incubation time was only 10 h, the depletion depends also on the growth parameters of the specific strains.
15) L301-309. The difference of B. and Ba. should be defined.
Comments on the Quality of English LanguageThere are some unclear sentences and typos.
Reviewer 3 Report
Comments and Suggestions for Authors
I reviewed the manuscript “Isolation and characterisation of Streptococcus spp. with HMOs utilization capacity from human milk” (foods-2951646). The manuscript idea is clearly presented, experimentation is described correctly, and results and discussion are appropriated.
Some minor comments:
I suggest no include abbreviations in the Tittle and Abstract sections.
Line 14. The word “oligosaccharides” is repeated twice in the same sentence, please check redaction.
Line 111 and118. Please include de dimension of the LC column.
Figure 1. Please check the image, there are some words underlined.
Round 2
Reviewer 2 Report
Comments and Suggestions for Authors
I sincerely thank the authors tof the modifications and clarifications. The manuscript has improved considerably.
Just as a remark. When naming N-linked sugars, the "N" should be in italics.
Comments on the Quality of English LanguageThe language is fine, even though the polishing can be done.